# *PEMT* rs7946 Polymorphism and Sex Modify the Effect of Adequate Dietary Choline Intake on the Risk of Hepatic Steatosis in Older Patients with Metabolic Disorders

**DOI:** 10.3390/nu15143211

**Published:** 2023-07-19

**Authors:** Chien-Hsien Wu, Ting-Yu Chang, Yen-Chu Chen, Rwei-Fen S. Huang

**Affiliations:** 1Ph.D. Program in Nutrition and Food Science, Fu Jen Catholic University, New Taipei City 242062, Taiwan; jasper5j@yahoo.com.tw; 2Department of Gastroenterology and Hepatology, Taipei Hospital, Ministry of Health and Welfare, New Taipei City 242033, Taiwan; 3Department of Nutritional Science, Fu Jen Catholic University, New Taipei City 242062, Taiwan; tingyoyo871123@gmail.com (T.-Y.C.); 94617643ad@gmail.com (Y.-C.C.)

**Keywords:** methyl donor nutrient, fatty liver, choline, *PEMT* polymorphism, rs7946

## Abstract

In humans, *PEMT* rs7946 polymorphism exerts sex-specific effects on choline requirement and hepatic steatosis (HS) risk. Few studies have explored the interaction effect of the *PEMT* rs7946 polymorphism and sex on the effect of adequate choline intake on HS risk. In this cross-sectional study, we investigated the association between *PEMT* polymorphism and adequate choline intake on HS risk. We enrolled 250 older patients with metabolic disorders with (*n* = 152) or without (*n* = 98; control) ultrasonically diagnosed HS. An elevated *PEMT* rs7946 A allele level was associated with a lower HS risk and body mass index in both men and women. Dietary choline intake—assessed using a semiquantitative food frequency questionnaire—was associated with reduced obesity in men only (*p* for trend < 0.05). ROC curve analysis revealed that the cutoff value of energy-adjusted choline intake for HS diagnosis was 448 mg/day in women (AUC: 0.62; 95% CI: 0.57–0.77) and 424 mg/day in men (AUC: 0.63, 95% CI: 0.57–0.76). In women, GG genotype and high choline intake (>448 mg/day) were associated with a 79% reduction in HS risk (adjusted OR: 0.21; 95% CI: 0.05–0.82); notably, GA or AA genotype was associated with a reduced HS risk regardless of choline intake (*p* < 0.05). In men, GG genotype and high choline intake (>424 mg/day) were associated with a 3.7-fold increase in HS risk (OR: 3.7; 95% CI: 1.19–11.9). Further adjustments for a high-density lipoprotein level and body mass index mitigated the effect of choline intake on HS risk. Current dietary choline intake may be inadequate for minimizing HS risk in postmenopausal Taiwanese women carrying the *PEMT* rs7946 GG genotype. Older men consuming more than the recommended amount of choline may have an increased risk of nonalcoholic fatty liver disease; this risk is mediated by a high-density lipoprotein level and obesity.

## 1. Introduction

Nonalcoholic fatty liver disease (NAFLD) is a common liver disease worldwide, with a prevalence of 20–40% in Western countries [1] and of 5–40% in Asia [2,3]. The progression of NAFLD is associated with increased rates of liver-related morbidity and mortality [4]. Recently, the prevalence of NAFLD has been increasing at an alarming rate [5]. Hepatic steatosis (HS), the early hallmark of NAFLD, results from an imbalance of dietary lipid intake, hepatic lipid metabolism, and lipid secretion by very-low-density lipoprotein (VLDL) from the liver [6]. The multiple-hit hypothesis of NAFLD development includes potential risk factors such as obesity, type 2 diabetes, dyslipidemia, metabolic syndromes, insulin resistance, and inflammation [7,8]. The interplay between dietary risk factors, such as a deficiency of methyl donor nutrients (MDNs; choline, betaine, and folate), and the aforementioned NAFLD-associated risk factors likely increases the risk of HS; however, the optimal amount of MDNs that should be consumed to minimize HS risk remains incompletely understood [9,10].

Choline is a key nutrient that plays pivotal roles in hepatic fat metabolism, methylation pathways, membrane phospholipid synthesis, and lipoprotein transport [11,12]. Phosphatidylcholine, a component of phospholipids in VLDL, is essential for the export of triglycerides from the liver [13]. Dietary choline intake ensures free choline moieties are available for the synthesis of phosphatidylcholine through the ATP-consuming cytidine diphosphate (CDP)–choline pathway, which satisfies 70% of the body’s phosphatidylcholine requirement [14]. The phosphatidylethanolamine *N*-methyltransferase gene (*PEMT*) encodes the enzyme phosphatidylethanolamine *N*-methyltransferase, which converts phosphatidylethanolamine into phosphatidylcholine in the liver through methylation; the methyl group is donated by S-adenosylmethionine, which is converted into S-adenosylhomocysteine. Through the activity of *PEMT*, 30% of the total phosphatidylcholine required by the body can be synthesized from phosphatidylethanolamine through methylation; for this process, other nutrients such as folate and betaine, in addition to S-adenosylmethionine, can also serve as methyl group donors [15]. Choline can be oxidized to betaine, which is a key osmolyte and serves as a methyl group donor in the epigenetic regulation of DNA [16,17]. Inadequate dietary choline intake impairs the CDP–choline pathway of phosphatidylcholine synthesis and the secretion of VLDL [18]. The endogenous synthesis of phosphatidylethanolamine through the *PEMT* pathway can only partially mitigate the negative effect of inadequate dietary choline intake on phosphatidylcholine synthesis. Animal studies have reported that low dietary choline intake or impaired endogenous choline biosynthesis, in addition to high-fat diet intake, reduces the choline level and increases fat accumulation in the liver [19,20]. Rats fed a diet deficient in choline and methionine developed HS and eventually NAFLD, which progressed to hepatocellular carcinoma [19]. Humans consuming a choline-deficient diet for 3 weeks exhibited a reduced choline pool and eventually developed liver dysfunction [11]. Choline is regarded as an essential nutrient, and adequate choline intake may reduce the risk of liver dysfunction [21]. However, the average daily requirement and recommended dietary allowance of choline to reduce the risk of HS remain to be determined.

Genetic polymorphisms of genes encoding enzymes that are associated with choline metabolism may considerably modify the choline requirement of individuals [22] and affect the efficiency of endogenous choline biosynthesis [23,24]. Common genetic variants of *PEMT*—rs7946 (+5465G→A) and rs12325817 (+744G→C) [25]—impair the de novo phosphatidylcholine synthesis pathway; this impairment has been associated with an increased risk of NAFLD [10,26]. Song et al. reported that the *PEMT* single nucleotide polymorphism (SNP) rs7946, a G-to-A substitution in exon 8, results in the substitution of valine with methionine, thereby inhibiting the activity of the resultant *PEMT* enzyme. AA homozygotes are more prevalent in patients with NAFLD than in healthy controls [10]. This SNP (rs7946) is extremely rare in the Japanese population. Nevertheless, the A allele is more prevalent in patients with nonalcoholic steatohepatitis than in healthy controls [27]. There is sexual dimorphism observed in the effect of *PEMT* polymorphism on NAFLD risk; this dimorphism may be attributed to the effect of estrogen [28], which binds to several estrogen response elements present in the promoter region of *PEMT* [28]. The genetic variants of *PEMT* are associated with not only HS risk in individuals with excessive calorie intake [20] but also NAFLD in lean patients [28]. *PEMT*−/− mice fed a choline-deficient diet exhibited significant reductions in their hepatic and plasma levels of phosphatidylcholine and the substantial hepatic accumulation of fat [29]. However, the condition could be reversed by feeding the mice a choline-supplemented diet [23]. Notably, *PEMT*−/− mice do not develop high-fat diet–induced obesity or insulin resistance, which indicates an interaction occurs between genetics and diet [30,31].

Few human studies have explored the influence of sex on how *PEMT* rs7946 polymorphism affects HS-associated metabolic markers to modify HS risk and whether adequate choline intake can minimize this genetic effect [25]. Therefore, in the current study, we aimed to investigate the effects of *PEMT* rs7946 polymorphism and sex on the association between dietary choline intake and NAFLD risk.

## 2. Materials and Methods

### 2.1. Study Participants

This cross-sectional, case-control study was conducted at the Department of Gastroenterology and Hepatology, Taipei Hospital (affiliated with the Taiwanese Ministry of Health and Welfare) between January 2020 and November 2020. Patients with hepatitis B or C infection, liver cirrhosis, severe cardiovascular disease, cancer, autoimmune hepatitis, or Wilson’s disease were excluded from this study. In addition, individuals with a weekly alcohol intake of >100 g and those whose alcohol intake data were unavailable were excluded. The study protocol was approved by the Joint Institutional Review Board of Taipei Hospital (approval number: IRB-0019-0021). Written informed consent was obtained from all participants after the face-to-face interview was completed.

HS was diagnosed by experienced physicians according to the Graif’s criteria by using an abdominal ultrasound scanner (Hitachi, Tokyo, Japan) [32]. The features of HS on abdominal ultrasonography are as follows: (1) diffuse enhancement of the near-field echo in the hepatic region, with a stronger extent of enhancement than that of enhancement in the kidney and spleen regions, and gradual attenuation of the far-field echo; (2) unclear display of intrahepatic lacuna structures; (3) mild to moderate hepatomegaly, characterized by a round and blunt border; (4) color Doppler ultrasonography showing a signal corresponding to reduced blood flow in the liver or a difficult-to-detect signal corresponding to a normal blood flow; and (5) unclear or nonintact display of the envelope of the right liver lobe and diaphragm. HS is graded as follows: absent (none of the aforementioned features are observed), mild (feature 1 and one of the following features: features 2–4), moderate (feature 1 and two of the following features: features 2–4), and severe (features 1 and 5 and two of the following features: features 2–4). In the present study, participants with an ultrasound-based diagnosis of moderate or severe HS were included in the HS group, and those without HS, as indicated by the normal echotexture of the liver on ultrasonography, were included in the control group.

Trained professionals conducted face-to-face interviews with all participants to obtain comprehensive medical and dietary data. A total of 255 individuals were determined to be eligible for participation in this study; of these patients, three failed to donate blood samples, and two failed to complete the questionnaire. The final HS and control groups comprised 152 and 98 individuals, respectively. Fasting blood samples were collected from all participants for biochemical measurement and for *PEMT* rs7946 polymorphism analysis.

### 2.2. Blood Biochemistry Marker Measurements

Within 1 week of their ultrasound-based HS diagnosis, blood samples were collected from the participants after they had fasted for at least 10 h. The plasma levels of glucose, insulin, total cholesterol, triglyceride, high-density lipoprotein (HDL) cholesterol, and low-density lipoprotein cholesterol were measured through enzyme-linked immunosorbent assay (Mercodia, Uppsala, Sweden) by using the Hitachi 911 analyzer. Furthermore, the plasma levels of aspartate transaminase, alanine transaminase, alkaline phosphatase, and gamma-glutamyl transferase were measured in accordance with the standard protocols (ITC Diagnostics, Taipei, Taiwan). Insulin resistance was evaluated using the homeostatic model assessment of insulin resistance (HOMA-IR) method; the HOMA-IR score was calculated by dividing the product of the fasting blood glucose level (mmol/L) and the fasting insulin level (mU/L) by 22.5 [33]. The fasting plasma folate level was measured using radioimmunoassay kits (Becton Dickinson, Franklin Lakes, NJ, USA), and the plasma homocysteine level was measured using a commercially available kit for fluorescence polarization immunoassay and the Abbott 130 AxSYM system (Becton Dickinson). The fasting plasma betaine and free choline levels were measured through liquid chromatography/electrospray ionization–isotope dilution mass spectrometry [9].

### 2.3. Assessment of Dietary Intake of MDNs

Dietary intake of folate, betaine, and choline was assessed using a specialized quantitative food frequency questionnaire (qFFQ-MDN), which was implemented using a previously described method [9,34]. The specialized qFFQ-MDN included more than 150 food items with high nutrient densities of folate, choline, and betaine, that is, 20 staple foods, 59 vegetables, 26 fruits, 38 meat and dairy products, 11 soybean products, and 9 types of nuts and fats. The food list included the 50 food items that are most frequently consumed that provide macronutrients and micronutrients, as indicated by the Nutrition and Health Survey in Taiwan database [35]. The qFFQ-MDN included the most frequently consumed food items that provide choline and betaine in Taiwan [36]. In the current study, the participants completed face-to-face interviews with registered dietitians and completed the qFFQ-MDN within 1 week after receiving an ultrasound-based HS diagnosis. The qFFQ-MDN was used to record each participant’s frequency of consuming standard servings of specific food items from five categories of food in the previous year; the participants reported their frequency of consumption per day, week, month, and year or indicated that they had never consumed a given food item. In addition, the participants were provided with visual aids, such as measuring cups and spoons, which they used to indicate the number of standard servings of each food item that they had consumed. The standard serving sizes in the qFFQ were based on the corresponding typical portion sizes consumed by the general population in Taiwan [35]. The questionnaire’s list of food items for evaluating folate, betaine, and choline levels was derived from the Nutrition and Health Survey in Taiwan database [35] and the US Department of Agriculture database [37], which were also used to assign MDN values to the food items. MDN values for nonidentical food items were assigned on the basis of those for similar food items. An analytical network and an app were used to generate a computerized nutrition database for calculating the total energy intake as well as the intake of macronutrients (carbohydrates, lipids, proteins, and fibers) and MDNs (folate, betaine, and choline). The qFFQ-MDN, database, analytical network, and app platform were registered at Fu Jen Catholic University Medical Research Center and maintained at the Biomedicine Core Research Lab at Fu Jen Catholic University.

### 2.4. Measurements of Body Composition, Body Fat Distribution, and Obesity

Anthropometric measurements were performed using a bioelectrical impedance whole-body composition analyzer (InBody 270 SELVAS Healthcare, Seoul, Republic of Korea). The following parameters were measured: whole-body fat, lean tissue mass, body fat percentage, waist-to-hip ratio, and visceral fat grade. Body mass index (BMI) was calculated by dividing body weight (kg) by height (m^2^). Visceral fat levels were graded according to the visceral fat tissue area (cm^2^) relative to the total body adiposity. On the basis of the criteria set by the World Health Organization, obesity was defined as having at least one of the following features: a BMI of >30 kg/m^2^, a body fat percentage of >25% for men and >32% for women, a waist-to-hip ratio of >0.9 for men and >0.85 for women, and a visceral fat level of >10 (visceral fat percentage > 10%) [38].

### 2.5. Determination of PEMT Genotypes

Real-time polymerase chain reaction (PCR) was performed to amplify a fragment including *PEMT* V175M polymorphism. The following primers were used for PCR: 5′-TGC TCC TGA CGG TGC TG-3′ (forward) and 5′-AGC GGT GAA GGG CTC TTC GTAT-3′ (reverse). The PCR fragment contained G-to-A substitution, which resulted in V175M amino acid substitution in the eighth exon of *PEMT* [39]. Probes were designed with consideration of point mutation sites and adjacent sequences. Probe 1 (5′-CAC CTA CAT AGT GGC TCT-3′) and probe 2 (5′-TCA CCT ACA TTA TGGCTC T-3′) were labeled using 6-carboxyfluorescein (FAM) and CAL560, respectively. When two probes are simultaneously hybridized on a single strand of DNA, fluorescence is released through the excitation and resonance effect of laser light. The FAM or CAL560 fluorescence emitted by a sample indicates whether the point mutation in *PEMT* is a G or an A allele. When FAM fluorescence is detected, both FAM and CAL560 fluorescence indicate the GA genotype and only CAL 560 fluorescence indicates the GG genotype. The quantitative PCR reaction was performed in a reaction mixture (volume, 20 μL) comprising 0.4 μL of *MTHFR* probe (10 μM forward primer, 10 μM reverse primer, and 10 μM probe), 300 ng DNA, and 10 μL of KAPA PROBE FAST qPCR Master Mix. The PCR protocol was as follows: initial denaturation at 95 °C for 3 min; 45 cycles of denaturation at 95 °C for 10 s, annealing at 58 °C for 40 s.

### 2.6. Statistical Analysis

Data were analyzed using SPSS (version 20.0; SPSS, Chicago, IL, USA) and R (version 4.2.1; R Foundation for Statistical Computing, Vienna, Austria). Continuous variables, presented as median and interquartile range (25–75%) values, were compared using the nonparametric Mann–Whitney U test. Trend analysis was performed using the Jonckheere–Terpstra test. Categorical variables were compared using the chi-square or Fisher exact test. The intake of macronutrients, including carbohydrates, lipids, and proteins, is presented as a percentage of total energy. The energy-adjusted dietary intake of MDNs, such as folate, choline, and betaine, was calculated using the residual method [40]. The optimal value of dietary choline intake for the diagnosis of HS was estimated based on a receiver operating characteristic (ROC) curve and the Youden index. *PEMT* polymorphism was evaluated using the Hardy–Weinberg equilibrium principle. Logistic regression models were constructed to identify the associations among the *PEMT* polymorphism, plasma choline level, and HS risk. Explanatory variables for HS risk, such as age, sex, BMI, blood lipid levels, and HOMO-IR scores, were incorporated into the models to adjust for the effects of various confounders. Restricted cubic spline (RCS) regression with four knots (0.10, 0.35, 0.65, and 0.90) was performed to investigate the nonlinear association between HS risk and dietary choline intake; this analysis was performed using R (version 4.2.1) [41,42]. The correlations among HS risk and other variables are presented as odds ratio (OR) and the corresponding 95% confidence interval (CI) values. A two-tailed *p* value of <0.05 was considered to indicate statistical significance.

## 3. Results

### 3.1. Participant Characteristics

Table 1 presents the demographic characteristics and biochemical measurements of the participants stratified by *PEMT* rs7946 polymorphism. The HS prevalence was significantly higher in women with the *PEMT* rs7946 GG genotype (61.8%) than in those with the GA (45.2%) or AA (26.8%) genotype. Similarly, the HS prevalence was significantly higher (75.9%) in men with the GG genotype than in those with the GA (63.2%) or AA (16.7%) genotype. Individuals (both men and women) with the GG genotype had the highest BMI, while those with the AA genotype had the lowest BMI. In women, the lipid profiles, including the levels of cholesterol, triglycerides, low-density lipoprotein, and HDL, did not vary significantly among individuals with the GG, GA, or AA genotype. However, men with the GG genotype had a higher triglyceride level, lower HDL level, and higher alanine transaminase level than did those with the GA or AA genotype; nonetheless, the between-group difference was nonsignificant. The HOMA-IR scores of individuals (both men and women) with the GG genotype were not significantly higher than those of individuals with the GA or AA genotype.

The plasma levels of the MDNs did not vary significantly among the individuals with the GG, GA, or AA genotype. The daily dietary intake of macronutrients and micronutrients, including MDN, was assessed using the qFFQ-MDN. Notably, in women, the energy-adjusted MDN intake varied nonsignificantly among individuals with the GG, GA, or AA genotype. The distribution of *PEMT* rs7946 polymorphism varied significantly between the HS and control groups (*p* < 0.001), with the GG genotype (71.1%) being more prevalent in the HS group than in the control group (48.0%). The HS group had a lower prevalence of the GA and AA genotypes (25.0% and 3.9%, respectively) than did the control group (31.6% and 20.4%, respectively; Appendix A). The frequency of the G allele was significantly higher in the HS group than in the control group. The genotype distribution in the study population followed the Hardy–Weinberg equilibrium. The mean BMI was significantly higher in the HS group than in the control group. Furthermore, the HS group had significantly higher levels of triglycerides, aspartate transaminase, alanine transaminase, gamma-glutamyl transferase, creatinine, and fasting glucose but a lower level of HDL than did the control group. The HS group also had a higher median insulin level, hemoglobin A1C level, and HOMA-IR score than did the control group (Appendix A).

### 3.2. Association of Energy-Adjusted Choline Intake with the Metabolic Markers of HS Varies Depending on PEMT rs7946 Polymorphism and Sex

Table 2 presents the risk of HS in the male and female participants stratified by total energy-adjusted dietary choline intake. In women with the GG genotype, the HS prevalence in the lowest choline intake tertile 1 was 82.6%, which was significantly higher (*p* = 0.023) than the prevalence in tertiles 2 (43.5%) and 3 (59.1%). By contrast, in men with the GG genotype, the HS prevalence was 62.1% in tertile 1, which was nonsignificantly lower (*p* = 0.104) than the prevalence in tertiles 2 (82.8%) and 3. The observed effect of the *PEMT* GG genotype on the association between choline intake and HS was independent of the degrees of HS-related metabolic markers, such as obesity, insulin resistance, and dyslipidemia, and the intake of other MDNs in both sexes. In women with *PEMT* rs7946 GA/AA genotype, higher tertiles of choline intake were not associated with a significantly lower risk of HS than lower tertiles were. In men with the GA or AA genotype, a higher tertile of choline intake was not associated with a significantly higher risk of HS (Table 2). In men and women, the prevalence of obesity did not vary between the tertiles of daily choline intake. Women with the GG genotype had slightly lower triglyceride levels, creatinine kinase (CPK) levels, hemoglobin A1C levels, and HOMA-IR scores in tertiles 2 and 3 than in tertile 1. The plasma levels of MDNs (folate, betaine, choline, and homocysteine) did not vary between the HS and control groups (Appendix A). In men with the *PEMT* rs7946 GG genotype, tertiles 2 and 3 of choline intake were associated with higher BMI values and HOMA-IR scores than were tertile 1, although this difference was nonsignificant (Appendix A).

### 3.3. Cutoff Values of Daily Dietary Choline Intake Influence the Risk of HS According to ROC Analysis

In women with *PEMT* rs7946 GG genotype, a higher tertile of energy-adjusted daily choline intake was significantly associated with a lower prevalence of HS. By contrast, in men with *PEMT* rs7946 GG genotype, a higher tertile of choline intake was insignificantly associated with a higher prevalence of HS. The cutoff value of energy-adjusted daily dietary choline intake for HS diagnosis was determined through an ROC analysis. For women with the GG genotype, the cutoff value of energy-adjusted choline intake for the diagnosis of HS was 448 mg/day (area under the curve [AUC]: 0.621, specificity: 0.769; sensitivity: 0.571; *p* = 0.098), and in tertiles 1 and 2 of energy-adjusted choline intake, the AUC value was 0.751 (specificity: 0.647; sensitivity: 0.828; *p* = 0.005). For men with the GG genotype, the cutoff value of energy-adjusted choline intake for the nondiagnosis of HS was determined to be 424 mg/day (AUC: 0.626; specificity: 0.571; sensitivity: 0.727; *p* = 0.083; Figure 1A,B). However, in individuals with the GA or AA genotype, the levels of choline intake and other MDNs were not significantly associated with HS. 

### 3.4. Interaction of PEMT rs7946 Polymorphism, Energy-Adjusted Daily Dietary Choline Intake, and Sex in Relation to HS Risk

HS risk was found to be influenced by sex, *PEMT* rs7946 polymorphism, and daily dietary choline intake. The participants’ biochemical characteristics were dependent on the interactions among sex, *PEMT* rs7946 polymorphism, and daily dietary choline intake (Figure 2). Elevated HOMA-IR scores, triglyceride levels, and BMI and reduced HDL levels were significantly associated with HS risk in women with the GG genotype or the GA or AA genotype and a dietary choline intake of <448 mg/day and that in men with the GG genotype and a dietary choline intake of <424 or >424 mg/day. In women with the GG genotype, a dietary choline intake of >448 mg/day, a high HOMA-IR score, and an elevated triglyceride level were significantly associated with HS.

Compared to an energy-adjusted daily dietary choline intake of <448 mg/day, an intake of >448 mg/day was associated with a significantly lower prevalence of HS in women with the *PEMT* GG genotype (80.0% vs. 47.4%, respectively; *p* = 0.011). Women with a dietary choline intake of >448 mg/day exhibited a nonsignificantly higher HDL level. Additionally, their BMI and CPK level were lower compared to those with a dietary choline intake of <448 mg/day (*p* = 0.056 and 0.009, respectively; Appendix A). In women, a dietary choline intake of >448 mg/day, combined with the GA or AA genotype, was associated with a lower HS risk (Table 3). Among women with the GG genotype, the odds ratio (OR) for HS was lower in individuals with a dietary choline intake of >448 mg/day compared to those with a dietary choline intake of <448 mg/day (OR: 0.194; 95% CI: 0.057–0.664). The GA or AA genotype and a dietary choline intake of >448 or <448 mg/day were associated with a significantly lower OR for HS compared to the GG genotype and a dietary choline intake of <448 mg/day. Even after adjusting for age, HOMA-IR score, triglyceride level, HDL level, and BMI, the OR for HS remained lower in women with the GA or AA genotype and a choline intake of >448 mg/day compared to those with the GG genotype and a dietary choline intake of <448 mg/day.

Compared to an energy-adjusted daily dietary choline intake of <424 mg/day, an intake of >424 mg/day was associated with an increased risk of HS in men with *PEMT* rs7946 GG genotype (OR: 3.556; 95% CI: 1.282–9.860). The increase in HS risk was not influenced by age, HOMA-IR score, or triglyceride level but it was affected by HDL level and BMI (Table 3).

### 3.5. PEMT rs7946 Polymorphism and Sex Regulate the Role of Adequate Choline Intake in Minimizing HS Risk

The association between HS risk and daily dietary choline intake was analyzed using RCS regression performed with four knots (0.05, 0.35, 0.65, and 0.95; Figure 3). The results revealed a nonlinear correlation between energy-adjusted dietary choline intake and HS risk; this correlation was further modified by sex and *PEMT* rs7946 polymorphism. Among women with *PEMT* rs7946 GG genotype, a U-shaped trend was observed in the association between dietary choline intake and HS risk. The linear and nonlinear *p* values of 0.016 and 0.041, respectively, indicated a decreasing trend in HS risk for increased choline intake in tertiles 1 and 2 (300–533 mg/day; Figure 3A). A dietary choline intake of >533 mg/day (tertile 3) mitigated the reduction in HS risk through adequate choline intake. After adjustments for age, triglyceride level, and HDL level, HS risk was found to be significantly higher in women with *PEMT* rs7946 GG genotype and a dietary choline intake of <448 mg/day (*p* = 0.044; Figure 4A). This finding suggests that there is an optimal range of adequate choline intake for minimizing HS risk in women with *PEMT* rs7946 GG genotype. However, in men with *PEMT* rs7946 GG genotype, HS risk increased with dietary choline intake, although nonsignificantly (Figure 3C). After adjustments for age, triglyceride level, and HDL level, HS risk was determined to be significantly higher in men with the GG genotype and a dietary choline intake of >424 mg/day (*p* = 0.0089; Figure 4B). However, dietary choline intake was not correlated with HS risk in women or men with *PEMT* rs7946 GA or AA genotype (Figure 3B,D).

## 4. Discussion

To the best of our knowledge, this cross-sectional, case–control study is the first to explore the effects of dietary choline intake on the risk of NAFLD with consideration of the effects of sex and *PEMT* rs7946 polymorphism. We determined the required daily dietary choline intake for women with *PEMT* rs7946 GG genotype for minimizing NAFLD risk. Choline and *PEMT* play pivotal roles in hepatic fat metabolism and the methylation pathways [11,12]. Genetic variations in MDN metabolism, such as *PEMT* rs7946 (+5465G→A) polymorphism, are associated with the risk and severity of NAFLD [45,46]. Studies in rats deficient in methionine and choline have demonstrated the development of hepatic steatosis (HS) with increased severity, reduced antioxidant response, and even hepatocellular damage [19]. In our study, the prevalence of the GG genotype was higher in the HS group compared to the control group (71.1% and 48%, respectively). By contrast, the prevalence of the AA genotype was lower in the HS group compared to the control group (3.9% and 20.4%, respectively). The distribution of *PEMT* rs7946 polymorphisms (GG, GA, and AA genotypes) in the HS and control groups showed variation between our study and studies conducted in other countries [46]. In the study by Song et al., the prevalence of the GG and AA genotypes in patients with fatty liver disease (*n* = 28; 7.1% and 67.9%, respectively) was lower and higher, respectively, compared to the control group (*n* = 59; 18.6% and 40.7%, respectively; *p* = 0.020) [10]. Song et al. also reported that individuals with *PEMT* rs7946 GA or AA genotype were more susceptible to liver damage due to insufficient choline intake (because of inhibited *PEMT* activity) compared to those with *PEMT* rs7946 GG genotype. However, the distribution of *PEMT* polymorphism was slightly different in the study by Romeo et al., who reported that the prevalence of the GG, GA, and AA genotypes was, respectively 29.4%, 34.7%, and 35.9% in patients with fatty liver disease (*n* = 170) and 24.8%, 45.9%, and 29.3% in the control group (*n* = 2052; *p* = 0.019) [47]. Dong et al. reported a high prevalence of the GG genotype both in Japanese patients with nonalcoholic steatohepatitis (*n* = 107; GG, 84.1%; GA, 7.5%; AA, 8.4%) and healthy controls (*n* = 113; GG, 95.6%; GA, 4.4%; AA, 0%) [27]. The finding of a higher prevalence of the GG genotype in patients with fatty liver disease compared to controls in our study differs from those of other studies as mentioned above. The variation in the distribution of *PEMT* rs7946 polymorphism across countries could be attributed to the ethnic differences among the study populations. The effect of *PEMT* polymorphism on NAFLD risk is influenced by various factors, including metabolic stress [20], dietary choline intake, host gut microbiota [48], and one-carbon metabolism.

The hepatic endogenous de novo choline synthesis pathway, which is dependent on the activity of *PEMT*, is crucial because it accounts for 20–30% of the total phosphatidylcholine synthesized in the body and can compensate for dietary choline deficiency [14]. The degree of *PEMT* activity has been reported to be 10–50% higher in female rats than in male rats [49]. Low hepatic *PEMT* activity in men and postmenopausal women [50] contributes to the risk of NAFLD in these individuals [45]. Notably, studies using *PEMT*-knockout mice have shown that choline deficiency can lead to the development of hepatic steatosis and liver injury, even when a diet supplemented with choline surpasses the required intake amount [51]. Men and women have different requirements for dietary choline and varying susceptibility to MDN deficiency [52,53]. Premenopausal women are less sensitive to liver injury-related choline deficiency than are men and postmenopausal women [21,54]. This is because estrogen enhances *PEMT* activity in the liver, leading to increased endogenous biosynthesis of choline, even during pregnancy and breastfeeding [28,45,54]. Most of the female participants in our study were postmenopausal women, making them susceptible to dietary choline deficiency. Sex, along with variations in lifestyle and nutrient status, can influence gene expression [55]. The sex-specific analyses performed in our study revealed genetic diversity in the interaction between genetics and nutrition in relation to the risk of NAFLD across individuals with different *PEMT* rs7946 variants. Moreover, the *PEMT* rs7946 variant may exhibit sexual dimorphism in terms of NAFLD risk or susceptibility. However, differences in the aforementioned interaction may be partially influenced by ethnicity and other yet undiscovered genetic factors. The interaction between genetics and environmental factors (e.g., geographic and socioeconomic factors) contributes to varying degrees of risk for NAFLD [56]. In our study, we discovered that the effect of the *PEMT* rs7946 GG genotype on HS risk differed between male and female participants. Among the women with the *PEMT* rs7946 GG genotype, the individuals with a higher dietary choline intake, particularly those consuming >448 mg/day, had a lower HS risk. Notably, the dietary choline requirement is higher for men and postmenopausal women than for premenopausal women [57]. In our study, >75% of the females participants were postmenopausal women with a low daily dietary choline intake which were associated with an increased HS risk in these women.

Our findings reveal that the effect of daily dietary choline intake on the risk of fatty liver disease is modified by *PEMT* rs7946 polymorphism and sex. After adjusting age, HOMA-IR score, triglyceride level, HDL level, and BMI were made for women with the GG genotype, individuals with a dietary choline intake of >448 mg/day were found to have a 79% lower HS risk than those with a dietary choline intake of <448 mg/day did (OR: 0.210; 95% CI: 0.054–0.819). Women with the GG wild-type genotype had an 81% lower HS risk than those with the GA or AA genotype did (OR: 0.19; 95% CI: 0.05–0.66); this lower risk was independent of choline intake and other HS-related metabolic factors, such as age, HOMO-IR score, dyslipidemia, and BMI. Women with the GG genotype had a higher triglyceride level, HOMA-IR score, and BMI but a lower HDL levels compared to those with the AA genotype (Table 1). A dietary choline intake of >448 mg/day was associated with a reduced risk of HS. Yue et al. reported an inverse association between high choline intake and NAFLD risk in middle-aged and older Chinese women with a normal body weight; this association was independent of the intake of other nutrients [58]. Fischer et al. demonstrated that the intake of a low-choline diet (<50 mg/day) for 6 weeks led to the development of fatty liver disease in 77% of all men, 80% of all postmenopausal women, and 44% of all premenopausal women [21]. The sexual dimorphism in NAFLD risk may be attributed to the effect of estrogen [28]. The efficiency of endogenous choline biosynthesis varies among individuals, with higher efficiency observed in premenopausal women compared to men and postmenopausal women due to the upregulation of *PEMT* expression by estrogen. However, a study revealed that reduced *PEMT* activity contributes to the risk of HS only in individuals with rapid triglyceride production, such as those with excessive calorie intake [20].

The pathogenesis of NAFLD is multifactorial and closely associated with metabolic stress. In individuals with the GA or AA genotype, the de novo choline synthesis pathway is suppressed and the CDP–choline pathway, which is dependent on energy, is activated to ensure an adequate supply of phosphatidylcholine [59]. The level of energy consumption is increased in individuals with the GA or AA genotype compared with that in those with the GG genotype. This increase in individuals with the GA or AA genotype may result in reduced metabolic stress, as evidenced by the finding of a low BMI and HOMA-IR score in this population in this study (Table 1). In *PEMT*-knockdown mice, *PEMT* deficiency was demonstrated to result in a significant reduction in the hepatic ratio of phosphatidylcholine to phosphatidylethanolamine in the case of obesity or overnutrition, which increases the risk of NAFLD [30]. Impaired *PEMT* activity is associated with inhibited weight gain and improved insulin sensitivity due to reduced endoplasmic reticulum stress [30]. Furthermore, impaired *PEMT* activity is associated with increased energy expenditure and reduced white-adipose lipogenesis. In our study, the majority of female participants were postmenopausal women who were susceptible to dietary choline deficiency. The protective role of adequate dietary choline intake against HS in women with the GA or AA genotype may become nonsignificant because they may experience reduced metabolic stress [21].

The effect of dietary choline intake on the risk of HS varies among men with the GG genotype, who are associated with obesity and increased metabolic stress, and men with the GA or AA genotype. After adjusting for age, HOMA-IR score, and triglyceride level in men with *PEMT* rs7946 GG genotype, individuals with an energy-adjusted dietary choline intake of >424 mg/day had a 3.5-fold increase in HS risk compared with the risk in those with an energy-adjusted dietary choline intake of <424 mg/day (OR: 3.556; 95% CI: 1.28–9.86). Further adjustments for HDL and BMI negated the significant effect of adequate choline intake (>424 mg/day) on HS risk. This suggests that HDL level mediates the risk of HS. Men with the GG genotype had a higher BMI, metabolic stress level, and insulin resistance level compared to women with the GG genotype, which might have negated the protective effects of an increased dietary choline intake on the risk of fatty liver disease. Excess dietary choline is metabolized by the gut microbiota to produce trimethylamine, which is then oxidized in the liver to form trimethylamine-N-oxide (TMAO). A high level of TMAO is associated with increased risks of cardiovascular disease and liver injury [60]. However, the thresholds for excessive choline intake and a harmful TMAO level remain under investigation. Notably, TMAO levels can be regulated by various factors, such as gut microbiota composition and kidney function. In our study, we did not measure TMAO levels and thus could not determine whether a dietary choline intake of >424 mg/day increases the risk of fatty liver disease in men with the GG genotype. Nonetheless, the cutoff value of 424 mg/day is lower than the recommended amount of choline intake for men in Taiwan (450 mg/day); thus, the risk of choline toxicity is minimal. Excessive dietary choline intake (600 mg/day) in women with the GG genotype may diminish choline’s protective role in mitigating the risk of HS (Figure 3A and Figure 4A). Further studies must be conducted to investigate the toxic effects of excessive dietary choline intake on HS risk and identify the dose–response relationship between the two parameters. Until more evidence is available, an excessive dietary choline intake must be avoided, particularly by individuals with a high level of metabolic stress.

Unlike in women with the GA or AA genotype, in men with the GA or AA genotype, increased dietary choline intake was not associated with an increased risk of fatty liver disease. In individuals with the GA or AA genotype and inadequate dietary choline intake, the risk of HS may be mitigated by increasing energy consumption and reducing metabolic stress, which is attributable to impaired *PEMT* activity. Compared to the GA or AA genotype, the GG wild-type genotype was associated with a slight although nonsignificant reduction in the risk of HS; this association was independent of other HS-related risk factors.

In Taiwan, a dietary choline intake of 450 mg/day is regarded as adequate for men, and that of 390 mg/day is regarded as adequate for women [61]. In the United States, these values are 550 and 425 mg/day for men and women, respectively [62]. Inadequate dietary choline intake has become a major concern in many countries [63]. For example, in Taiwan, <5% of women consume the recommended amount of choline (390 mg/day) [36]. We recommend a dietary choline intake of 448 mg/day to reduce HS risk in women; this amount is significantly higher than the adequate intake amount and the average daily choline intake in real-world scenarios [36]. *PEMT* polymorphism can impair the activity of the *PEMT* enzyme and reduce the pool of free choline, thus exacerbating the effect of dietary choline deficiency [10]. However, we discovered that *PEMT* GA or AA genotype was not associated with an increased risk of HS, even in individuals with low choline intake. Few studies have focused on the complex interaction between genetics and nutrition. The genetic factors contributing to HS risk are multifactorial and may involve multiple genes [64]. Common *PEMT* polymorphisms, such as rs7946, rs12325817, rs4646343, rs3760188, rs1531100, and rs4646365, affect choline metabolism and impair the de novo phosphatidylcholine synthesis pathway, which increase individuals’ susceptibility to liver damage due to insufficient dietary choline intake [10,22,65]. Given that a single genetic variant is limited in its ability to explain the risk of NAFLD, the development of a polygenic risk score may be a more favorable approach [64]. A polygenic risk score is the sum of all trait-associated alleles carried by an individual and represents the individual’s genetic predisposition to develop a disease or an associated outcome [66]. Because the risk of multiple gene-associated fatty liver disease can be gradually mitigated through diet and lifestyle modifications [67,68], future studies may focus on developing personalized nutritional interventions that are designed with consideration of each individual’s genetic predisposition to NAFLD.

Metabolic factors are closely associated with the risk of NAFLD. In our study, both male and female participants with NAFLD had a significantly elevated BMI, triglyceride level, CPK level, and HOMA-IR score but a reduced HDL level. *PEMT* rs7946 polymorphism was associated with certain metabolic risk factors, particularly those associated with metabolic stress. Because most participants were enrolled from clinics and had prevalent metabolic factors or chronic diseases, the effect of *PEMT* rs7946 polymorphism on HS risk might have been mitigated by other risk factors. Furthermore, we discovered that dietary choline intake was significantly associated with the intake of other MDNs, such as betaine and folate. We previously demonstrated that low dietary MDN intake is associated with visceral adipose accumulation, which affects the risk and progression of NAFLD [9]. In our study, the level of free choline in plasma was higher in the HS group than in the control group. However, this level may not necessarily be a positive indicator of dietary choline intake. High plasma levels of free choline and high blood levels of homocysteine have been associated with an increased risk of hepatic fat accumulation in patients with NAFLD [69]. Future studies are warranted to investigate the correlation between the plasma level of phosphatidylcholine and the risk of NAFLD.

## 5. Strengths and Limitations

To the best of our knowledge, this is the first study to focus on the correlations between dietary choline intake, sexual dimorphism, and *PEMT* rs7946 polymorphism and their effects on the risk of NAFLD. We have identified an association between dietary choline intake and HS risk by evaluating the effects of sex and *PEMT* rs7946 polymorphism. The participants’ MDN intake was evaluated and their blood biomarkers were assessed to analyze the interaction between nutrient intake and genetics. Our findings indicate the necessity for personalized dietary choline intake recommendations for women with *PEMT* rs7946 polymorphism.

This case-control study has several strengths. Firstly, to the best of our knowledge, this study is the first to explore the effects of sex and *PEMT* polymorphism on the amount of dietary choline required to minimize NAFLD risk; we focused on women with *PEMT* rs7946 GG genotype. Secondly, we investigated the risk factors for HS in a high-risk population burdened with metabolic disorders. Finally, our findings indicate a distinct association is present between dietary choline intake and NAFLD risk in individuals stratified by sex and the *PEMT* rs7946 polymorphism.

This study has several limitations. Firstly, although the control group received no HS diagnosis, the participants in the control group were not completely healthy. The participants were enrolled from clinics and were mostly older adults with multiple metabolic diseases that were being treated. Consequently, our findings may not be generalizable to all patients with NAFLD because of likely selection biases involving age and comorbid metabolic diseases or factors. Secondly, HS was diagnosed using liver ultrasonography. However, the gold standard for HS diagnosis is histological analysis of liver tissues obtained through biopsy. Biopsy is the only means of acquiring histological samples for HS diagnosis and staging. However, biopsy is an invasive method and may be substituted with ultrasonography, which is a convenient and cost-effective method for examining patients with suspected HS without using radiation. A meta-analysis reported that ultrasonography had a sensitivity of 85% and specificity of 94% for HS diagnosis [70]. However, ultrasonography cannot provide precise or objective data because of potential inter-operator variation. In this study, the HS group included patients who had received a diagnosis of only moderate HS; this was ensured to avoid any possible operator-dependent diagnosis bias. Thirdly, the participants’ dietary intake history was recorded based on recall data; therefore, the intake amounts of the nutrients might have been underestimated, particularly because of the advanced age of the participants. Furthermore, dietary history data were collected using the qFFQ only at participant enrollment; recall bias might have introduced inaccuracies into the calculation of choline intake amounts. The participants’ choline intake amounts were analyzed on the basis of hypothetical food composition data obtained from the US Department of Agriculture database, which may not accurately correspond to Taiwanese choline-containing food items; this problem was addressed by this study using the validated qFFQ [34]. Fourthly, the candidate gene approach adopted in this study has certain shortcomings, including selection and publication biases and poor replicability. Fifthly, our study had a cross-sectional design; therefore, we could not establish any causal relationship between nutrients and genetics in the pathogenesis of NAFLD. Sixthly, a substantial amount of evidence indicates that multiple genetic factors contribute to the development of NAFLD [64]. However, we analyzed the effect of only a single *PEMT* rs7946 polymorphism, which might have been influenced by other genetic factors. Finally, the sample size was insufficiently large for a comprehensive and adequately powered analysis to be conducted. This limitation was particularly notable with respect to the stratification of the participants by sex, gene polymorphism, and dietary choline intake. In the future, we will increase the sample size to appropriately address the effects of sex and genetic diversity. In addition, we will explore other MDNs associated with metabolic pathways that contribute to the risk and progression of NAFLD.

## 6. Conclusions

The present study provides compelling evidence suggesting that a dietary choline intake of >448 mg/day is associated with a lower risk of hepatic fat accumulation in women with *PEMT* rs7946 GG genotype, regardless of age, BMI, triglyceride level, HDL level, and HOMA-IR score. However, excessive dietary choline intake may be associated with a higher risk of HS in men with *PEMT* rs7946 GG genotype. The effect of dietary choline intake on HS risk is modified by sex and *PEMT* rs7946 polymorphism. Therefore, the recommended amount of dietary choline intake for minimizing HS risk should be adjusted, taking into consideration of polymorphism in individuals’ choline-related genes. This study may inspire further research on MDNs and nutrient-related gene polymorphisms to develop effective dietary interventions for fatty liver disease.

## Figures and Tables

**Figure 1 nutrients-15-03211-f001:**
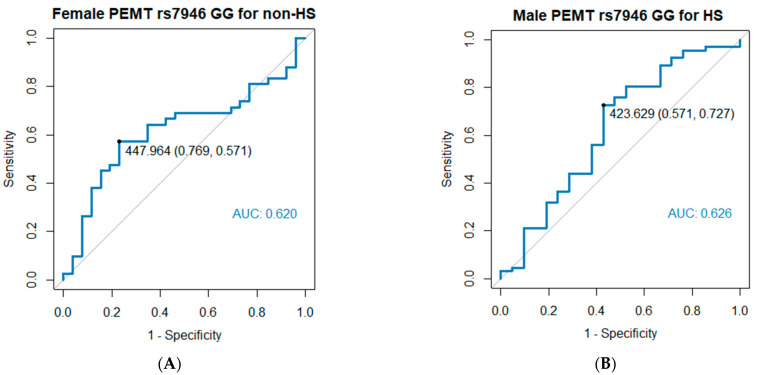
Cutoff values of daily dietary choline intake that influence HS risk. (**A**) Women with the rs7946 GG genotype. The cutoff value of energy-adjusted daily choline intake for the diagnosis of non-HS was 448 mg/day. (**B**) Men with the rs7946 GG genotype. The cutoff value of energy-adjusted daily choline intake for the nondiagnosis of HS was 424 mg/day. HS, hepatic steatosis.

**Figure 2 nutrients-15-03211-f002:**
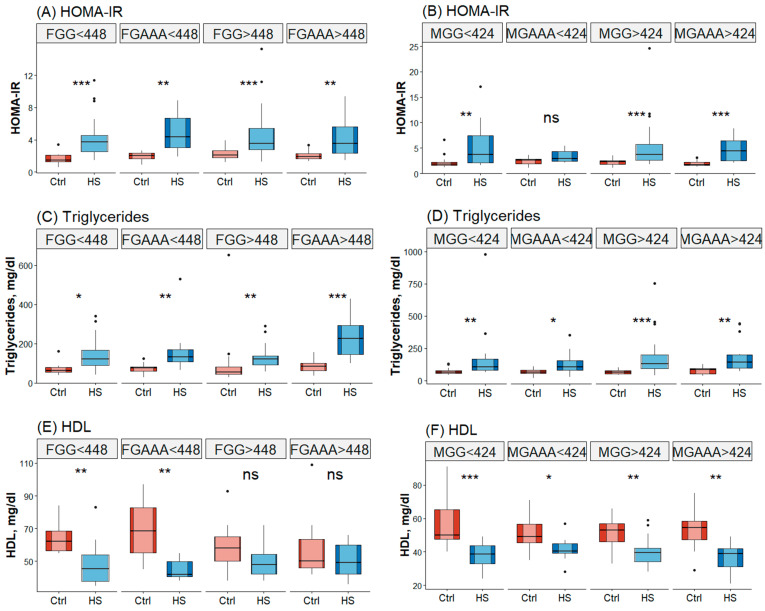
Biochemical characteristics of the male and female participants stratified by *PEMT* rs7946 polymorphism and choline intake. FGG: women with the *PEMT* rs7946 GG genotype; FGAAA: women with the *PEMT* rs7946 GA or AA genotype; <448: a dietary choline intake of <448 mg/day; >448: a dietary choline intake of >448 mg/day; MGG: men with the *PEMT* rs7946 GG genotype; MGAAA: men with the *PEMT* rs7946 GA or AA genotype; <424: a dietary choline intake of <424 mg/day; >424: a dietary choline intake of >424 mg/day. * *p* < 0.05; ** *p* < 0.01; *** *p* < 0.005; **** *p* < 0.001; and ns, nonsignificant.

**Figure 3 nutrients-15-03211-f003:**
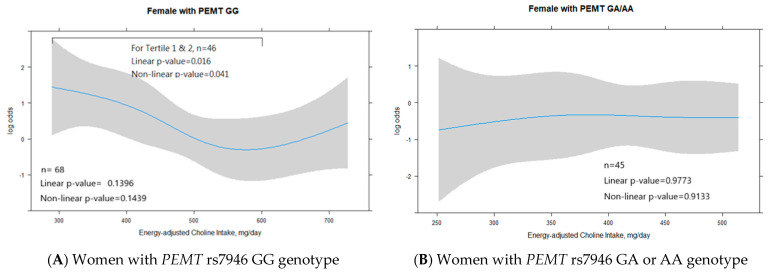
Association of HS risk and energy-adjusted dietary choline intake according to sex and *PEMT* rs7946 polymorphism. With consideration of the effects of sex and *PEMT* rs7946 polymorphism, we conducted restricted cubic spline regression to identify the nonlinear relationship between HS risk and energy-adjusted dietary choline intake; four knots were used in the restricted cubic spline regression: 0.05, 0.35, 0.65, and 0.95. HS refers to hepatic steatosis.

**Figure 4 nutrients-15-03211-f004:**
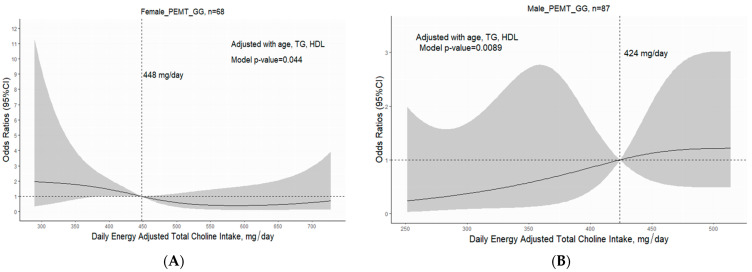
Odds ratio for HS and energy-adjusted dietary choline intake in *PEMT* rs7946 GG genotype. Data were analyzed through restricted cubic spline regression with consideration of the effects of sex and *PEMT* rs7946 polymorphism. (**A**) Women with *PEMT* rs7946 GG genotype (reference: a dietary choline intake of 448 mg/day). (**B**) Men with *PEMT* rs7946 GG genotype (reference: a dietary choline intake of 424 mg/day). HS refers to hepatic steatosis.

**Table 1 nutrients-15-03211-t001:** Levels of choline-associated methyl donor nutrients, hepatic steatosis-related metabolic markers, and body adiposity in participants stratified by *PEMT* rs7946 polymorphism and sex.

Characteristics	GG	GA	AA	GG vs. GA	GG vs. AA	GG vs. GA/AA	*p* for Trend
Women (*n* = 113)	68 (60.2)	31 (27.4)	14 (12.4)	*p*-Value	*p*-Value	*p*-Value	
*n* (%)	68 (60.2)	31 (27.4)	14 (12.4)				
Age, yrs	64 (56–69)	59 (56–67)	60.5 (56.8–66.0)	0.233	0.234	0.135	0.155
BMI, kg/m^2^	24.9 (22.6–28.0)	23.6 (22.2–26.5)	22.4 (19.6–25.3)	0.123	0.047	0.032	0.020
Hepatic steatosis	42 (61.8)	14 (45.2)	4 (28.6)	0.122	0.023	0.023	0.013
Obesity *	54 (79.5)	24 (77.4)	8 (57.1)	0.822	0.077	0.311	0.127
Cholesterol (mg/dL)	199 (172–224)	196 (183–231)	209 (173–244)	0.544	0.618	0.478	0.467
Triglycerides (md/dL)	98.5 (62.3–138)	112 (77–156)	83 (57.5–125)	0.264	0.562	0.554	0.784
HDL (mg/dL)	51 (42.3–62.0)	48 (42–60)	60.5 (50.5–78.3)	0.841	0.016	0.321	0.138
AST (U/L)	26 (21.0–32.8)	24 (21–32)	26 (19.0–37.0)	0.528	0.81	0.545	0.593
ALT (U/L)	27 (19.0–42.8)	21 (18–38)	24 (17.3–40.5)	0.210	0.426	0.176	0.204
HOMA-IR	2.78 (1.97–4.05)	2.36 (1.77–3.53)	2.13 (1.71–2.81)	0.346	0.156	0.159	0.159
Choline, μmol/L	10.5 (9.03–12.8)	10.7 (9.38–13.1)	10.3 (9.33–11.9)	0.870	0.699	0.955	0.883
Plasma Hcy, μmol/L	9.18 (7.66–11.3)	9.45 (7.06–12.2)	8.56 (6.24–12.6)	0.877	0.382	0.591	0.532
Energy, kcal	1320 (1105–1864)	1348 (1115–1623)	1569 (1037–2140)	0.411	0.605	0.694	0.905
Folate, μg	647 (465–935)	761 (568–1110)	492 (341–1070)	0.880	0.355	0.376	0.69
Betaine, mg	215 (151–318)	250 (191–395)	227 (160–322)	0.085	0.786	0.143	0.229
Choline intake, mg	468 (383–557)	499 (394–638)	414 (342–541)	0.419	0.388	0.828	0.909
Men (*n* = 137)							
*n*	87 (63.5)	38 (27.7)	12 (8.8)				
Age, yrs	59 (49–65)	62.5 (48.5–68.0)	60.5 (54.0–68.0)	0.317	0.306	0.207	0.186
BMI, kg/m^2^	27.2 (23.8–30.8)	25.8 (22.8–28.2)	24.2 (22.6–26.8)	0.102	0.020	0.020	0.011
Hepatic steatosis	66 (75.9)	24 (63.2)	2 (16.7)	0.146	<0.001	0.004	<0.001
Obesity	63 (72.4)	18 (47.4)	6 (50.0)	0.007	0.113	0.004	<0.011
Cholesterol (mg/dL)	193 (165–210)	181 (166–200)	199 (157–226)	0.153	0.772	0.284	0.381
Triglycerides (md/dL)	105 (71.0–165)	91.5 (62.8–146)	82.5 (62.8–120)	0.288	0.214	0.100	0.127
HDL (mg/dL)	41.0 (35.0–48.0)	45.0 (39.5–51.0)	46.5 (37.8–62.3)	0.217	0.109	0.088	0.065
AST (U/L)	26 (21–36)	25 (18–36)	25 (20.5–33.0)	0.207	0.679	0.221	0.252
ALT (U/L)	32 (22–52)	24.5 (19.0–57.3)	22.5 (18.5–37.8)	0.168	0.189	0.090	0.085
HOMA-IR	3.23 (2.14–5.42)	2.74 (2.04–4.25)	2.53 (2.13–2.77)	0.158	0.118	0.067	0.054
Choline, μmol/L	13.1 (10.9–15.5)	12.3 (10.4–14.9)	10.9 (10.8–14.6)	0.316	0.336	0.217	0.196
Plasma Hcy, μmol/L	12.7 (10.5–15.2)	12.7 (10.6–15.2)	12.7 (11.3–17.2)	0.788	0.447	0.589	0.476
Energy, kcal	1819 (1337–2174)	1750 (1411–2174)	1637 (1365–2168)	0.797	0.94	0.806	0.813
Folate, μg	577 (416–810)	476 (362–746)	460 (320–859)	0.127	0.319	0.092	0.099
Betaine, mg	206 (144–307)	181 (133–235)	297 (119–349)	0.106	0.855	0.204	0.288
Total choline, mg	475 (395–577)	462 (342–519)	382 (323–447)	0.308	0.015	0.062	0.036

The intake of macronutrients was adjusted for energy by using the density method [43]. The intake of one-carbon nutrients was adjusted for energy by using the residual method [44]. Discrete variables are presented in terms of numbers and percentages and were compared using the chi-square test. Between-group differences were considered to be statistically significant at *p* < 0.05. Continuous variables are presented in terms of median and interquartile range values and were compared using the Mann–Whitney U test. Trend analysis was performed using the Jonckheere–Terpstra test. Between-group differences were considered to be statistically significant at *p* < 0.05. * Obesity was defined as the presence of at least one of the following features: a BMI of >30 kg/m^2^, a waist-to-hip ratio of >0.85 for women and >0.9 for men, a visceral fat level (InBody test) of >10, and a body fat percentage of >32% for women and 25% for men. Leanness was defined as the absence of all of the aforementioned features. GG: *PEMT* rs7946 GG genotype; GA: the *PEMT* rs7946 GA genotype; AA: the *PEMT* rs7946 AA genotype; GA/AA: GA or AA genotype.

**Table 2 nutrients-15-03211-t002:** Associations of energy-adjusted dietary choline intake with hepatic steatosis and obesity.

Choline Intake	Tertile 1	Tertile 2	Tertile 3	*p*-Value	*p*-Trend
Female, *PEMT* rs7946 GG (*n* = 68)					
Choline, mg/day	213–420	420–533	533–914		
Number	23	23	22		
NAFLD	19 (82.6)	10 (43.5)	13 (59.1)	0.023	0.101
Obesity *	21 (91.3)	16 (69.6)	17 (77.3)	0.181	0.241
Female, *PEMT* rs7946 GA/AA (*n* = 45)					
Choline, mg/day	170–404	404–528	528–982		
Number	15	15	15		
NAFLD	6 (40.0)	7 (46.7)	5 (33.3)	0.757	0.712
Obesity *	10 (66.7)	12 (80.0)	10 (66.7)	0.649	1
Male, *PEMT* rs7946 GG (*n* = 87)					
Choline, mg/day	219–418	418–506	506–865		
Number	29	29	29		
NAFLD	18 (62.1)	24 (82.8)	24 (82.8)	0.104	0.067
Obesity *	11 (37.9)	5 (17.2)	5 (17.2)		
Male, *PEMT* rs7946 GA/AA (*n* = 68)					
Choline, mg/day	134–369	369–473	473–867		
Number	17	16	17		
NAFLD	10 (58.8)	7 (43.8)	9 (52.9)	0.684	0.734
Obesity *	8 (47.1)	9 (56.2)	7 (41.2)	0.684	0.734

Associations of energy-adjusted dietary choline intake with hepatic steatosis risk and obesity were modified by sex and *PEMT* polymorphism. Discrete variables are presented in terms of numbers and percentages and were compared using the chi-square test. Between-group differences were considered to be statistically significant at *p* < 0.05. * Obesity was defined as the presence of at least one of the following features: a BMI of >30 kg/m^2^, a waist-to-hip ratio of >0.85 for women and >0.9 for men, a visceral fat level (InBody test) of >10, and a body fat percentage of >32% for women and 25% for men. Leanness was defined as the absence of all of the aforementioned features. Dietary choline intake was adjusted for total energy by using the residual method [44]. GG: *PEMT* rs7946 GG genotype; GA: the *PEMT* rs7946 GA genotype; AA: the *PEMT* rs7946 AA genotype; GA/AA: GA or AA genotype.

**Table 3 nutrients-15-03211-t003:** Sex-specific effects of *PEMT* rs7946 polymorphism and choline intake on HS risk.

*PEMT* rs7946	*PEMT* GG	*PEMT* GA/AA	*PEMT* GG	*PEMT* GA/AA	
Female					
Intake, mg/day	Choline < 448	Choline < 448	Choline > 448	Choline > 448	
HS (*n*)	24	8	18	10	
Control (*n*)	7	12	19	15	
Odds Ratio (95% CI) Ref					*p*-int
Model 0	1	0.194 (0.057–0.664)	0.276 (0.096–0.798)	0.194 (0.061–0.621)	0.115
Model I	1	0.125 (0.027–0.584)	0.177 (0.048–0.658)	0.113 (0.026–0.494)	0.233
Model II	1	0.154 (0.032–0.755)	0.193 (0.050–0.739)	0.124 (0.027–0.558)	0.263
Model III	1	0.191 (0.038–0.970)	0.210 (0.054–0.819)	0.136 (0.029–0.627)	0.343
Male					
Intake, mg/day	Choline < 424	Choline < 424	Choline > 424	Choline > 424	
HS (*n*)	18	12	48	14	
Control (*n*)	12	12	9	12	
Odds Ratio (95% CI) Ref					*p*-int
Model 0	1	0.667 (0.226–1.970)	3.556 (1.282–9.860)	0.778 (0.269–2.250)	0.124
Model I	1	1.246 (0.343–4.534)	3.778 (1.190–11.994)	0.799 (0.225–2.837)	0.230
Model II	1	0.959 (0.229–4.017)	3.094 (0.833–11.490)	0.534 (0.127–2.252)	0.161
Model III	1	0.719 (0.163–3.166)	2.017 (0.508–8.006)	0.456 (0.103–2.029)	0.309

Model 0 was not adjusted for any covariate. Model I was adjusted for age, HOMA-IR score, and triglyceride level. Model II was adjusted for age, HOMA-IR score, triglyceride level, and HDL level. Model III was adjusted for age, HOMA-IR score, triglyceride level, HDL level, and body mass index. Data are presented in terms of OR (95% CI) values. *p*-int is the *p*-interaction value for the interaction between choline intake and *PEMT* rs7946 polymorphism in women (448 mg/day) and men (424 mg/day). Between-group differences were significant at *p* < 0.05. Abbreviations: CI, confidence interval; HDL, high-density lipoprotein; HOMA-IR, homeostatic model assessment of insulin resistance; HS, hepatic steatosis; OR, odds ratio; *p*-int: *p*-interaction.

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
