# Peer review of "PEMT rs7946 Polymorphism and Sex Modify the Effect of Adequate Dietary Choline Intake on the Risk of Hepatic Steatosis in Older Patients with Metabolic Disorders"

_nutrients, 2023, doi:10.3390/nu15143211_

Round 1

Reviewer 1 Report

The paper "Fermented Protaetia brevitarsis Larvae Ameliorate Chronic Ethanol-Induced Liver Injury in Mice through AMPK and TLR-4/NF-κB Signaling Pathways" deals on the combined effect of PEMT rs7946 polymorphism and sex on the dietary choline intake and related risk of hepatic steatosis in older patients with metabolic disorders. The paper is well written and experiments and results convincing and figures of good quality. Only a minor remark is given to the Authors, to make the paper suitable for publication.

Minor remark- Tables 1, 2; Figure 2: not mandatory, but at the convenience of the reader the addition of the meaning of AA, GA, GG to the legends would be useful, as anyway done for the other Figures.  

good

Author Response

Minor remark- Tables 1, 2; Figure 2: not mandatory, but at the convenience of the reader the addition of the meaning of AA, GA, GG to the legends would be useful, as anyway done for the other Figures. 

Answers: In order to provide a more comprehensive reading experience for the readers, we have added the meaning of AA, GA, and GG to the bottom of the legends in Tables 1, 2, and Figure 2. Additionally, we have included additional explanations in other tables to enhance understanding.

Reviewer 2 Report

The topic of this article is clear, rational, rich content and strong innovation. However, some minor issues still need to be improved. After minor revision, it can be accepted.

1) The author should check the whole manuscript for any grammatical errors and any other differences. Writing needs considerable improvement. 

2) Please pay attention to the font format in Table 3.

3) Please pay attention to the correct citation of references.

Minor editing of English language required

Author Response

1) The author should check the whole manuscript for any grammatical errors and any other differences. Writing needs considerable improvement.

Answers: We reviewed the whole manuscript again and completed minor editing of the text for better presentation, as indicated by the marked changes. We have made grammatical adjustments throughout the document, reinforcing the clarity and precision of the content. We have also attached the revised manuscript along with the previous “English editing certificate”.

2) Please pay attention to the font format in Table 3.

Answers: We made minor adjustments to the font format of Table 3 to improve clarity in its presentation.

3) Please pay attention to the correct citation of references.

Answer: We have reviewed the manuscript and references. Additionally, we have made 8 minor text edits, updated 4 referenced papers, and included one additional reference paper, as indicated by the marked changes.
